# Study on Multiple Fractal Analysis and Response Characteristics of Acoustic Emission Signals from Goaf Rock Bodies

**DOI:** 10.3390/s22072746

**Published:** 2022-04-02

**Authors:** Xuebin Xie, Shaoqian Li, Jiang Guo

**Affiliations:** School of Resources and Safety Engineering, Central South University, Changsha 410083, China; xbxie@csu.edu.cn (X.X.); 205512137@csu.edu.cn (S.L.)

**Keywords:** mine goaf, acoustic emission monitoring, fractal theory, wavelet noise reduction, multiple fractals, response characteristics

## Abstract

Based on the actual monitoring data of the acoustic emission (AE) ground pressure monitoring and positioning system, this paper introduces fractal theory and the multifractal detrended fluctuation analysis (MF-DFA) method to estimate the waveform multifractal spectrum of goaf rock acoustic emission signals and investigate multifractal time-varying response characteristics. The research results show that the wavelet hard thresholding method has the best noise reduction effect on the original signal, and the box counting dimension has a strong waveform identification effect. Before deformation damage occurs, fractal spectral width Δ*α* shows an increase and then decrease and the fluctuation scale factor Δ*f*(*α*) decreases and then increases. When damage occurs, fractal spectral width Δ*α* decreases and then stabilizes and concentrates. Simultaneously, the fluctuation scale factor Δ*f*(*α*) keeps decreasing until the lowest point, and then shows an increasing trend until it reaches a stable state. This study is of great significance to the stability evaluation and disaster early warning of mine goaf.

## 1. Introduction

The development and application of mineral resources is the cornerstone of modern social development. However, the use of mining methods such as the room-and-pillar mining method and the shrinkage method has left a large number of goafs in the mine area. Underground mining operations break the original rock stress state of the quarry, causing the redistribution of stresses in the surrounding rock near the quarry. The legacy of the mining goaf makes the surrounding pillars subject to more complex ground stresses [1]. When the design of the pillar in the mining area is not reasonable, and the load on the pillar exceeds its own ultimate strength, disastrous accidents such as rock misalignment and mining area collapse occur [2,3]. The usage of cut-and-fill stopping can attenuate the hazards of mined areas, but higher filling costs limit it, and there are fewer application cases in small and medium-sized mines. The accumulation of these constraints means that many mines fail to treat the generated goaf areas in a timely manner, and the number and volume of goaf in underground mines are expanding [4]. Therefore, it is of great practical significance and social value to carry out long-term monitoring and stability evaluation of underground goaf zones [5].

From a deep viewpoint, the nature of rock damage is the macroscopic manifestation of internal microcracks breaking, expanding and then penetrating to produce cracks. In 1950, the German scientist Kaiser found that when the stress in polycrystalline metals was released from its historical maximum level and then reloaded, very little acoustic emission was generated, while a large amount of acoustic emission was generated when the stress reached and exceeded the historical maximum level [6]. Many scholars have demonstrated through experiments that rock materials also have a significant Kaiser effect. For example, Liu et al. [7] carried out Brazilian disc acoustic emission experiments on granite and marble to investigate the relationship between acoustic emission properties and microscopic rupture mechanisms under cleavage loading. Meng et al. [8] studied the energy accumulation and evolution characteristics of 30 sandstone specimens under six different loading rates and analyzed the stress-strain relationship and AE characteristics to reveal the energy evolution law of rock deformation damage. These studies link acoustic emission phenomena closely with rock destabilization and damage, providing new channels to determine the stability of rock masses.

Considering the energy release and transformation perspective, the expansion and fracture of rock microcracks results in part of the internally stored energy released being in the form of stress waves, which can be monitoring instruments in form of acoustic emission signals. The stress state of the surrounding rock body in the underground mining goaf is constantly changing, so the energy release is also continuously changing without interruption, which allows the AE signal to convey a variety of information [9]. The AE signal has rich characteristic parameters, such as ringing count, large events number, event rate, energy rate, and others. Studying the variation law of AE characteristic parameters can provide guidance for practical engineering. The practice of several mines has shown [10,11,12] that AE monitoring technology can be used to monitor rock engineering sites such as mining hollow areas that are difficult to be accessed by technicians. The stability state of rock bodies in mining hollow areas can be judged by the change characteristics of AE signals.

Fractal theory is an important branch of nonlinear scientific research that addresses numerous unsmooth and irregular geometries in nature and describes the degree of geometry irregularity by fractal dimension. By nature, the AE signal is a complex set of nonlinear and non-smooth time series. Many scholars have studied the signal characteristics of AE time series by fractal theory and obtained landmark research results. Xie et al. [13,14] took the lead in applying fractal geometry theory to the rock mechanics field and systematically studied the correlation between fractal geometry and rock fracture, forming a more complete system of fractal rock mechanics theory. Yuan et al. [15] used the acoustic emission acquisition and analysis system and the box counting method to calculate the fractal dimension of AE spatial distribution. The fractal dimension showed a decreasing trend with the increase of the load intensity. Zhang et al. [16] studied the deformation and AE characteristics of gas coal rocks under true triaxial loading conditions and analyzed the variation pattern and AE fractal characteristics under different gas pressures and surrounding pressures. The results showed that the AE correlation dimension can accurately reflect the damage evolution process of coal samples. In conclusion, fractal theory provides a new perspective for the study of rock acoustic emissions and rock damage [17,18].

Multifractals were introduced by Grassberger [19] in the 1980s. Multifractal theory describes the local singularities of measures and functions in terms of geometric probabilities. It uses generalized information dimensions and multifractal spectra to describe fractal objects and develops important directions in fractal theory [20]. Fractals and rescaled extreme difference analysis can only provide a holistic and average description of the object under a data characteristics study, which makes it difficult to reflect the differences between data structure levels. Compared to single fractal methods, multiple fractal methods describe the fractal structure through spectral functions that can describe the volatility of the fractal object at different levels in a more refined way [21]. Therefore, deconstructing the AE signal with multiple fractals can determine the crack evolution process inside the rock at a deeper level, and even capture the precursor information of rock damage.

Cai et al. [22] used multifractals to deconstruct and analyze AE signal characteristics and found that the AE multifractal spectral patterns at different loading stages were characterized by small probability events, which illustrated the complexity of the rock damage process. Kong et al. [23] used multiple fractal theory to structure the electromagnetic signal naturally generated by coal. The preliminary study showed that the multiple fractal spectrum of the electromagnetic signal is characterized by small probability events, and the characteristic parameters of the electromagnetic radiation signal in the later stages of coal heating are very different from those in the earlier stages of heating. Kong et al. [24] conducted loading tests on gas-bearing coal rocks. The results showed that the width of the multifractal analysis spectrum was smaller in the low stress phase and increased in the high stress phase. These studies and findings fully illustrate the application of the multiple analysis method in rock acoustic emission signal processing.

In this study, a large amount of data was collected by an acoustic emission monitoring and positioning system, and then the wavelet threshold noise reduction method was used to reduce the noise of the obtained AE waveforms. The fractal dimension was applied to accurately identify different kinds of waveforms generated in the underground mining area. Finally, the multifractal spectrum and other characteristic parameters of the AE waveform were estimated based on multifractal theory. This study reveals the multifractal time-varying response characteristics of the AE waveforms during deformation and damage of the rock body in the mining area, which provides important early warning information for the safe production of the mine. The main purpose of this study is to establish a rock instability early warning method in goaf based on multifractal theory through a large amount of acoustic emission signal data, so as to provide a guarantee for mine production safety.

## 2. Acoustic Emission Monitoring System

### 2.1. Engineering Background

A mine in the Guangxi Zhuang Autonomous Region, China, is mainly mined currently using the shallow hole retention method [25]. The mine area is a low mountainous hilly landscape with a highland in the middle extending in a north-east-east direction. The highest elevation in the mine area is 115 m and the lowest elevation is 55 m, with a relative height difference of 60 m. The ore body is located below the local erosion datum, and the main water-filled aquifer is more developed in the shallow karst, with medium water-richness. The deep karst is weakly developed and poorly water-rich, but the karst water has higher water pressure, more complicated hydrogeological boundary conditions, and may produce problems such as karst ground collapse and ground subsidence and deformation. At present, the mine is equipped with production stopes from −170 m in the middle section to −320 m in the middle section. The stope distribution space is large, forming a situation of simultaneous mining of multiple middle sections. Coexisting multiple sections of the mining area interact with each other to form a multi-layered complex mining area group with wide distribution and large volume scale. With the further extension of the mining area to the deep, the ground pressure of the multi-layered stacked mining area group will become more complex.

### 2.2. Layout of Monitoring System

Unlike most indoor rock mechanics test data, the AE signal data in this study are derived from field measurements in the underground mining area. The acoustic emission monitoring system uses STL-24 acoustic emission ground pressure monitoring equipment, as shown in Figure 1. The acoustic emission monitoring system as a whole is divided into the ground monitoring and analysis center, the −120 m mid-range data acquisition and exchange center and the −270 m mid-range data acquisition and exchange center.

The ground monitoring and analysis center is located at the dispatch station in the mine office area. The −120 m mid-section data acquisition and exchange center is located in the pump room in the −120 m mid-section underground and the −270 m mid-section data acquisition and exchange center is located in the dedicated monitoring chamber in the −270 m mid-section underground. The sensors are positioned in the mining tunnel or inter-pillar of the quarry where they can be easily installed and where noise interference is weak, to reduce the impact of mining operations. The fiber optics are placed at a suitable height and close to the tunnel wall, with special signs to remind the operator to protect the fiber optic lines. The actual installation is shown in Figure 2.

The STL-24 multi-channel acoustic emission monitoring and positioning system has a highly reliable central processing unit. The system adopts an A/D board as the analog-to-digital conversion interface that can achieve 24 channels of high-speed synchronization to achieve overrun or delayed trigger. The system has great ability to suppress field noise, a friendly operation interface, and an intuitive image.

### 2.3. Data Acquisition

The mine was mined by the shallow hole retention method in the early stage, leaving a large number of empty areas that have not been filled over time. There have been many destabilization instances of the surrounding rock body, as shown in Figure 3. In this study, acoustic emission monitoring probes were arranged near the main gobs in the well. All monitoring probes were connected to the corresponding underground data acquisition center through optical fiber, and the lower computer of the underground data acquisition center transmitted the acoustic emission data to the ground monitoring center. The monitoring personnel could observe and analyze the goaf acoustic emission signal through the upper computer of the ground monitoring center. During the operation of the acoustic emission monitoring system, this study carried out the daily monitoring, monthly report analysis and stope stability evaluation of the surrounding rock mass in the goaf. This provides an important reference for production safety decisions.

## 3. Theory and Method

### 3.1. Wavelet Denoising

Acoustic emission data obtained at engineering sites often contain a large amount of abnormal noise, and waveform noise reduction is an important step in processing the original waveform. Wavelet denoising is a typical time-frequency domain signal processing technology, similar to short-time Fourier transform and empirical mode decomposition (EMD) methods. The comparison of these methods is shown in Table 1 below. In most cases, the traditional Fourier transform can be applied to separate the noise in the signal only when the frequency band overlap between signal and noise is small. If the frequency bands overlap too much, this method cannot achieve the effect of signal-to-noise separation through filtering [26]. The wavelet transform overcomes this deficiency. This uses different properties of the decomposed noise signal and the useful signal to perform a series of transforms on the wavelet coefficients to achieve the signal-to-noise separation effect. At present, there are many denoising methods related to wavelet transform [27], such as the Lipschitz exponent method and the threshold noise reduction method. These methods all promote the development of wavelet transform in signal denoising. In this study, the wavelet threshold noise reduction method was used to preprocess the obtained original acoustic emission waveform. The basic flow chart is shown in Figure 4.

We assumed that the signal with noise can be expressed as *f*(*t*) = *s*(*t*) + *n*(*t*), where *s*(*t*) represents the original letter and *n*(*t*) represents the white Gaussian noise, and follows N (0, *σ*^2^) normal distribution. Wavelet transform is a kind of linear transform, in which part of the wavelet coefficients after discrete transformation is the signal, and part is the noise. The signal and the noise exhibit different characteristics after being decomposed. Signals often correspond to coefficients with relatively large amplitudes, while noise corresponds to relatively small amplitudes. It is precisely by using this difference that the decomposed coefficients can be processed by selecting an appropriate threshold. The coefficients larger than the threshold can be regarded as generated by the signal, and the coefficients smaller than the threshold can be regarded as generated by noise. After a series of operations such as elimination and retention, the remaining coefficients can be reconstructed to obtain the signal after noise removal.

After noise reduction of the acoustic emission signal, the noise reduction effect was evaluated by two indicators: signal-to-noise ratio (SNR) and root mean square error (RMSE). SNR refers to the ratio of pure signal to noise in an acoustic emission signal. In general, a larger SNR means higher signal quality and less noise. The calculation method is shown in Equation (1):(1)$${\mathrm{SNR}}_{db}=10\log_{10}\left(\frac{P_{signal}}{P_{noise}}\right)$$

Usually, the acquired acoustic emission signal is discrete, and the power can be directly calculated according to the time series of the signal, as shown in Equation (2):(2)$$P_{signal}=\frac{1}{n}\sum_{k=1}^{n}s_{k}^{2}$$

The RMSE is the square root of the ratio of the square of the deviation of the predicted value from the true value to the number of observations *n*. The RMSE is used to measure the deviation between the observed value and the true value. The smaller the RMSE, the higher the accuracy of the data. The calculation method is as Equation (3):(3)$$\mathrm{RMSE}=\sqrt{\frac{1}{n}\sum_{i=1}^{m}w_{i}{\left(y_{i}-{\hat{y}}_{i}\right)}^{2}}$$

### 3.2. Waveform Recognition

The next process after signal noise reduction processing is waveform identification. The complex environment of underground mines, and the wide variety of signals collected by acoustic emission monitoring systems, make it necessary to accurately classify acoustic emission events. Fractal theory has obvious advantages and has more applications in the field of waveform recognition [28]. In general, traditional mechanical theories and studies are based on classical integer dimensional space, while fractal theory extends the geometric measure of space to the fractional dimension. The principle of self-similarity is one of the most important principles of fractal theory [29]. This characterizes the fractal as scale-independent under geometric transformation. Fractal theory is broadly divided into two categories. One is the fractal that strictly satisfies the self-similarity condition, such as the Koch curve, Peano curve and so on. The other is the fractal characteristic within a certain range, and the fractal characteristic may disappear beyond that range. The time signal sequences are not visually self-similar but have statistical self-similarity. Although the apparent similarity may not be visible from the acoustic emission signal profile and local amplification, their statistical parameters are consistent. In this case, the fractal dimension remains constant as the curve is amplified [30]. This fully proves the statistical self-similarity of acoustic emission signals.

A fractal is irregular, fractional and fragmented. It is a graph, phenomenon or physical process with self-similar characteristics that can be described by the fractal dimension. There are many methods to calculate the fractal dimension [31,32,33,34], such as Hausdorff dimension, similarity dimension, box-counting dimension and capacity dimension. In this study, the Hausdorff dimension and box-counting dimension were mainly used for description.

#### 3.2.1. Method of Hausdorff Dimension

The Hausdorff dimension was proposed by the German mathematician Felix Hausdorff as a method that can accurately measure complex sets dimensionality [35]. Suppose there exists a non-empty set *U* in n-dimensional Euclid space and the diameter of the set *U* is:(4)|U|=sup{|x−y|:x,y∈U}
where *y* is the coordinate of any two points in the set *U*, i.e., the diameter of *U* is the maximum distance between any two points. If there exists an infinite number of such sets, for all *i*:(5)$$0<\left|U_{i}\right|<\delta ,X\subset \overset{\infty }{\underset{i=1}{\cup }}U_{i}$$

Those satisfying the above conditions are said to be *δ*-covered by *X*.

For ∀δ>0, $X\subset R^{n}$ and *D* ≥ 0 define:(6)$$H_{\delta }^{D}(X)=\inf\left\{\sum_{i=1}^{\infty }{\left|U_{i}\right|}^{D}\right\}$$

It can be easily deduced from the above definition that as *δ* decreases, the *δ*-covering of *X* also decreases gradually. Therefore, when *δ* tends to 0, we have:(7)$$H^{D}(X)=\underset{\delta \to 0}{\lim}H_{\delta }^{D}(X)=\sup{\sum_{j}^{\infty }\left|U_{j}\right|}^{D}$$
where *H^D^*(*X*) is the *D*-dimensional Hausdorff measure of *X*. If there exists *t* > *D*, then:(8)$$\sum_{i}{\left|U_{i}\right|}^{t}\leq \delta^{t-D}\sum_{i}{\left|U_{i}\right|}^{D}$$

When there is *δ* tend to 0 and *H^D^*(*X*) < ∞, then there exists *D**_cr_* such that the following equation holds:(9)$$H^{D}(X)=\left\{\begin{array}{l} \infty ,D<D_{cr}\\ 0,D>D_{cr} \end{array}\right\}$$

*D**_cr_* is the Hausdorff dimension of *X*, denoted as dim *X* = *D**_cr_*

#### 3.2.2. Box Counting Dimension Method

The box counting dimension is a fractal object covered with a lattice of different sizes, and the fractal dimension is calculated by computing the relationship between the intersecting boxes number and the size of the sizes. Compared with other fractal dimensions, the physical meaning of the box counting dimension is clearer and the calculation is easier to implement. It has a large number of applications in many fields [36,37,38].

The acoustic emission signal *S*(*t*) is dual-scale, with the time scale *δ*_1_(*ms*) in the transverse direction and the vibration amplitude *δ_2_*(*mv*) in the longitudinal direction. Assume that the time course curve *L* ∈ *R*^2^ of the acoustic emission signal divides *R* × *R* into the smallest possible grid *kδ*_1_ × *kδ*_2_ (*k* = 1,2,3,…, which indicates the magnification of the grid). Assuming that the curve box dimension is the sum of the number of grids intersecting the acoustic emission waveform curve L as *N*_*kδ*_1__ or *N*_*kδ*_2__, the box counting dimension *D_B_* can be defined as:(10)$$D_{B}=\underset{\begin{array}{l} \delta_{1}\to 0\\ \delta_{2}\to 0 \end{array}}{\lim}\frac{\mathrm{lg}N_{k\delta_{i}}}{-\mathrm{lg}k\delta_{i}}$$

### 3.3. Multifractal Analysis

A multifractal is an infinite number collection of scalar indices defined on a fractal structure. Multiple fractals describe the overall characteristics from its localization with spectral functions and investigate the distribution probability measures pattern of characteristic covariates by means of statistical physics. Since multifractals involve structural and hierarchical features of time series data, multifractals can identify subtle differences between data [39]. In this study, multifractal detrended fluctuation analysis (MF-DFA) was used to calculate the multifractal characteristics of acoustic emission signals.

The salient feature of the method is that it makes full use of the sequence data length by dividing the acoustic emission sequences in equal length from the forward and reverse bi-directional order [40]. In addition to this, the method uses least squares to fit a polynomial for each segment, thus eliminating the effect of non-smooth trends in the time series. MF-DFA uses different order fluctuation functions to analyze the scalar behavior of the time series at different levels, fine out the fractal characteristics of the inscribed time series and reveal the multiple fractal characteristics hidden in the non-stationary time series. The computational flow of MF-DFA is shown in Figure 5.

The specific calculation steps are as follows. Step 1: Given the acoustic emission time series *x*(*t*), the length of the series is *N* and the mean value of the series is $\bar{x}$, on this basis, find the cumulative error series *y*(*t*) of *x*(*t*) with respect to $\bar{x}$:(11)$$y(t)=\sum_{i=1}^{t}\left(x(i)-\bar{x}\right)$$

Step 2: Given a time scale *s*, the sequence *y*(*t*) is divided into equal parts using *s* as the criterion. In total, it can be divided into *m* consecutive and non-overlapping subintervals of equal length, where *m* = int(*N*/*s*). The inverse order method is used to make full use of the data information, and the previous operation is repeated from the end of the sequence, resulting in 2*m* subintervals.

Step 3: The trend is fitted to each subinterval, and the trend part is subtracted from the original base to obtain the corresponding residual series, which is denoted as z*_v_*(*t*):(12)$$z_{v}(t)=y_{v}(t)-p_{v}^{k}(t)$$
where: $y_{v}(t)$ is a subinterval, $p_{v}^{k}(t)$ is a polynomial of order *k* fitted to the *v*th subinterval. *v* takes values in the range [1, 2*m*] and *t* takes values in the range [1, *s*].

Step 4: Calculate the mean squared deviation of the residual series *z_v_*(*t*):(13)$$F^{2}(s,v)=\frac{1}{s}{\sum_{t=1}^{s}\left[z_{v}(t)\right]}^{2}$$

Step 5: Take the mean value of the data set and calculate the *q*-order volatility function of the time series:(14)Fq(s)={12m∑v=12m[F2(s,v)]q2}1q

The above steps lead to the *q*-order fluctuation function *F**_q_*(*s*) corresponding to a certain scale *s*. Transforming the values of *s*, the above steps are repeated to obtain a series of point values of *s*-*F**_q_*(*s*). If there is a long-range correlation in this time series, the following power-law relationship between *F**_q_*(*s*) and *s* is obtained:(15)$$F_{q}(s)\propto s^{h(q)}$$

Taking the logarithm of both sides of the above equation, we get:(16)$$\mathrm{lg}F_{q}(s)=h(q)\mathrm{lg}s+\mathrm{lg}b$$
where *h*(*q*) is the corresponding generalized Hurst index and *b* is a constant factor.

If *h*(*q*) is a fixed constant, this means that the sequence has a single fractal character and no multifractal character. If *h*(*q*) is a nonlinear decreasing function of *q*, this means that the sequence has multifractal features [41]. The intensity of fractal features and fractal singularity of the acoustic emission time series can be characterized by the multifractal spectrum, which is calculated as follows.
(17)$$\tau \left(q\right)=qh(q)-1,\alpha =\tau^{\prime }(q),f\left(\alpha \right)=q\alpha -\tau \left(q\right)$$
where *τ*(*q*) is denoted as the Renyi index, also known as the scalar function. *α* is the singular intensity and *f*(*α*) is the multifractal spectrum.

If the scalar function is a nonlinear up-convex function of *q*, this means that the acoustic emission time series has multiple fractal characteristics, or if the scalar function is a linear function, this means that the acoustic emission time series exhibits single fractal characteristics. According to this principle, the curve characteristics of the scalar function can be used to determine whether the acoustic emission time series has multifractal characteristics. When the *α*-*f*(*α*) curve shows a single-peaked convex shape and resembles a quadratic function, this means that the time series has multiple fractal characteristics. When *α*-*f*(*α*) converges to a single point, the time series is unifractal.

In addition, the fractal spectral width Δ*α* and the fluctuation scale factor Δ*f*(*α*) are used to quantitatively describe the multiple fractal properties of the acoustic emission time series. Δ*α* can reflect the singularity of the acoustic emission signal and the spatial differences in the evolution of the properties, which in turn can be linked to the energy release process inside the rock. The larger the Δ*α*, the greater the multifractal intensity of the waveform. Δ*f*(*α*) mainly reflects the proportion of small and large fluctuations in the waveform. The larger the Δ*f*(*α*), the greater the proportion of small fluctuations in the waveform. The equations for the parameters are shown below:(18)$$\Delta \alpha =\alpha_{\max}-\alpha_{\min},\;\Delta f(\alpha )=\Delta f(\alpha_{\max})-\Delta f(\alpha_{\min})$$

## 4. Results

### 4.1. Wavelet Threshold Denoising

Wavelet thresholding noise reduction processing methods are generally classified into three types: hard thresholding, soft thresholding and fixed thresholding. Taking the acoustic emission signal A as an example, the original acoustic emission signal is processed for noise reduction using hard, soft and fixed threshold distributions with MATLAB software tools as summarized in Figure 6. It can be seen that the amplitude of the fixed-threshold processed waveform is dramatically reduced compared with the original waveform, and the waveform tends to be a fixed smooth curve, discarding a large number of peaks and valleys. The overall amplitude of the waveform after soft thresholding is reduced, and the oscillation becomes weaker and tends to be smoother. The waveform after hard threshold processing is staggered, and the original amplitude range is retained to the greatest extent, but some local segments are also discarded.

To determine the optimal wavelet noise reduction method, 30 sets of data were randomly selected to find the optimal noise reduction method based on all 90 sets of acoustic emission data. The SNR and RMSE were used as evaluation indexes, and the calculation results are shown in Figure 7. It can be seen that the wavelet hard-threshold noise reduction method has the highest SNR and the lowest RMSE. The comparison results of evaluation indexes show that the acoustic emission signal of this mine is suitable for the wavelet hard threshold noise reduction method to complete the noise reduction process of the waveform.

### 4.2. Waveform Classification

Underground mining operations take many different forms and, therefore, produce different types of signals, such as mechanical noise signals generated by the mobile equipment and vibration signals generated by rock drilling and blasting operations. The fluctuations generated by these factors are captured by monitoring instruments and thus reflected in the waveforms. The correct identification of rock acoustic emission signals and the exclusion of noise source acoustic emission signals are important for the analysis of ground pressure activities and stability evaluation studies in mine mining areas [42]. In long-term monitoring, the main signals collected by the monitoring system are the surrounding rock body waveform, shovel operation waveform, rock drilling operation waveform and the blasting operation waveform. These four typical underground mine acoustic emission signal waveforms are shown in Figure 8.

From the performance of a large number of waveforms, it was found that different sources produce different types of waveforms, which in turn lead to waveforms with different parametric characteristics, as shown in Table 2. The blasting operation waveform has a faster decay effect and a larger average amplitude because the blasting occurs for a short period of time. Mining mobile machinery such as scrapers produce faster signal frequencies, less dense waveforms, and smoother curves. The surrounding rock body waveform has an obvious peak period, with a large waveform rise and fall in a certain period of time, and then returns to normal. In general, the amplitude of the surrounding rock body waveform is lower than that of other noise waveforms.

### 4.3. Fractal Dimension Calculation

From the perspective of fractal theory, different kinds of waveforms also have different performances in terms of trend changes of fractal dimension, and classification and identification of waveforms can be achieved according to this feature [43]. Analysis by only one fractal dimension does not fully reflect the change of different kinds of waveforms, so this study used the Hausdorff dimension and box-counting dimension together to analyze the change pattern of different waveforms.

For each waveform, 30 waveforms were randomly selected, and the Hausdorff dimension and the corresponding box counting dimension of each waveform were calculated, as shown in Figure 9. The fractal dimension of the blasting operation waveform varies from 1.4 to 1.65. The fractal dimension of the shovel operation waveform varies smoothly and is distributed in the interval of 1.5 to 1.6. The fractal dimension of the surrounding rock body waveform fluctuates sharply, and the extreme difference reaches 0.57. The fractal dimension of the rock drilling operation waveform also fluctuates more, second only to the surrounding rock body waveform, and the extreme difference reaches 0.35.

From the box counting dimension calculation results, it can be seen that the four different waveforms show completely different box dimension data characteristics, among which the rock drilling operation waveform has the largest fractal dimension, followed by the surfacing rock body waveform, blasting operation waveform and shovel operation waveform in order. The rock drilling operation waveform has the largest fractal dimension, followed by the surfacing rock body waveform, blasting operation waveform, and shovel operation waveform, and the rock drilling operation waveform has the most concentrated waveform distribution with high average energy distribution, which leads to the largest box counting dimension. The box counting dimension of the surrounding rock body waveform is closer to that of the blasting operation waveform and has a crossover region. However, the average value of the surrounding rock body waveform box counting dimension is higher than the blasting operation waveform.

A comparison of the Hausdorff dimension and box counting dimension of the four waveforms shows that the Hausdorff dimension fluctuates to a greater extent and has a greater extreme difference value. In the Hausdorff dimension, the surrounding rock body waveform has the highest mean value. Among the box counting dimensions, the rock drilling operation waveform has the highest mean fractal dimension. The box counting dimension of each waveform is more obviously stratified than the Hausdorff dimension, and the rock drilling operation waveform and blasting operation waveform are more similar in the Hausdorff dimension, which is less discernible. In summary, the results obtained by using box counting dimension are clearer and more discriminative. The box-count dimension can be used as one of the important bases for waveform classification and identification.

### 4.4. Multifractal Analysis

#### 4.4.1. Key Parameter Setting

The collected rock acoustic emission signal is a complex nonlinear and non-smooth time series. Rock rupture in goaf mining areas is often characterized by a discontinuous multiscale, and the multiple fractal methods can describe the different levels of volatility of such rock rupture signals more finely than the simple fractal dimension.

Based on the monitoring data, multiple fractal analysis was performed for the measurement points and the multivariate measurement point series composed of measurement points, respectively, where the fluctuation order q was taken to be in the range [–10, 10]. The lg*Fq*(*s*)-lg*s* bilogarithmic scatter plot was made and fitted by the least squares method, and its slope was the generalized Hurst exponent *h*(*q*). Taking a certain group of the surrounding rock body waveforms as an example, the analysis results are shown in Figure 10. It can be clearly seen that the curve is better fitted when lg*s* = 2.94~2.97. Special attention is paid to the fact that the local Hurst index *h*(*q*) with a large time length shows a smooth and slow trend, which is related to the smaller number of larger calculation intervals. Combined with Equation (15), the slope of the curve fit is the generalized Hurst index *h*(*q*), which can be used for subsequent estimation of the multifractal spectrum. At this time, the multiple fractal time length *s* is taken as: *s*_min_ = 870, *s*_max_ = 934.

#### 4.4.2. Multiple Fractal Spectrum

Figure 11 shows the variation of the generalized Hurst exponent and the scalar function *τ*(*q*) for the measurement point as a whole and the individual measurement points, respectively. When *q* varies between [–10, 10], the generalized Hurst exponent is not constant for either the measurement point as a whole or each measurement point component sequence, but shows a non-linear decreasing trend with the change of *q*. This indicates that the displacement sequences from the measurement point, as a whole, to each internal measurement point have obvious multifractal characteristics, and the description by single fractal theory alone is inadequate. At different *q* fluctuation orders, the overall generalized Hurst index curve analysis concentrates on the lower fluctuation of each single measurement point, which exhibits better multifractality. The *h*(*q*) values of either the overall or the fractal are still significantly larger than 0, indicating that the signal measurement series has good memory and long-range correlation from the overall to the local fractal, which are both non-stationary and stochastic.

When *q* < 0, *h*(*q*) is mainly influenced by the small fluctuation variance, and when *q* > 0, *h*(*q*) is mainly influenced by the large fluctuation variance. As can be seen from Figure 10, when *q* < 0, the *h*(*q*) values distribution of each measurement point is more discrete, and the differences in the strength of positive persistence are obvious. When *q* > 0, the *h*(*q*) values consistency of each measurement point is good, and they have similar positive persistence. Furthermore, the *h*(*q*) values difference between measurement points is smaller when *q* > 0 than when *q* < 0. This explains that under the influence of large fluctuation factors such as water level, the decreasing trend of each measurement point is basically the same, the overall change trend is also the same, and the area where the measurement points are located maintains a more stable long-range correlation.

On the *h*(*q*) basis, the scalar function *τ*(*q*) is calculated by Equation (17). It can be concluded that the scalar function consistency of each measurement point is good, and the middle part is up-convex. The *τ*(*q*) value is −1 when q is taken as 0, and the overall nonlinear relationship further confirms that each displacement measurement point has multiple fractal characteristics.

Based on this, the multifractal spectrum of the signal is calculated as shown in Figure 11. The multifractal spectrum image has a single-peaked convex distribution and resembles a quadratic function curve. The local scale of the displacement multifractal is not constant, which portrays the diversity of local variations at different moments. The singular intensity *α* is mainly concentrated on both sides of the image, which reflects the uneven distribution of the displacement sequence fractal structure. From the results, the overall multiple fractal spectrum is basically symmetrical and has good synergy, which indicates that the acoustic emission signal develops in a stable state. From Figure 12, the Δ*α* of the signal is 7.038 and the Δ*f*(*α*) is 0.036, which indicates that the multifractal intensity is larger he fluctuations are more intense and complex, and the proportion of small fluctuations in the waveform is smaller.

### 4.5. Multifractal Time-Varying Response Characteristics of Waveform

Destabilization damage in underground mining areas is caused by the sprouting, expansion and the interaction of microfractures within the rock mass, which is a non-smooth dynamic evolutionary process in a time series [44]. Therefore, it is necessary to focus on analyzing the multiple fractal features variation characteristics of rock microfracture acoustic emission waveforms with time. Then, combined with the macroscopic damage law at the site, the interrelationship between the acoustic emission waveform multifractal characteristics and the damage at the site is found, which provides a theoretical basis for the establishment of the early warning method for the destabilization of the mining area.

On 6 March 2017, a localized fall occurred in the west gang pillar of No. 622-① in the −220 middle section of the mine, with a fall area of 25–30 m^2^ and about 1300 tons of crushed stone. This study used the acoustic emission monitoring data from 7 February to 8 March 2017 to start the analysis and research. At this time, the TD16 monitoring channel was the closest to the site of the fall, so the data from the TD16 monitoring channel were used. Figure 13 shows the variation pattern of the multiple fractal spectral parameters Δ*α* and Δ*f*(*α*) of the rock body acoustic emission waveform near the TD16 monitoring channel with time.

The upper limit value of Δ*α* and the lower limit value of Δ*f*(*α*) have very obvious characteristics of time-series change, so the evolution trend of Δ*α* is taken as the upper limit value of the acoustic emission event group, and the evolution trend of Δ*f*(*α*) is taken as the lower limit value of the acoustic emission event group. As can be seen in Figure 12, the fluctuations from 7 February to 21 February are very sharp, mostly concentrated in the interval range of 0−2. However, there are also more data points scattered and distributed within the interval of 2−8, and the average distance between the data is larger. Δ*α* shows a sharp increasing trend at the beginning, reaching a maximum of 8.01, indicating that the intensity of the multiple fractal increases and the fluctuations become complex and violent. Overall, it experienced a process from low to high to low, forming a clear wave. At the same time, Δ*f*(*α*) shows a decreasing trend, reaching a minimum of −1.86, indicating an increase in the proportion of large fluctuations in the waveform time series. On the whole, it experiences the development process of a wave trough from high to low to high, and the data are concentrated and distributed between −1.0 and 0.5, with a more discrete distribution area. This is due to the fact that the strong unloading effect of underground excavation obstructs the further expansion of microcracks inside the hard rock block, and the number of induced acoustic emission events increases. At this moment, the local stress is highly concentrated, and the strain energy stored inside the rock increases, which is in the calm period before the deformation damage occurs.

After 21 February, Δ*α* decreased overall and the distribution was more concentrated with a mean value of 0.76 and in a stable state, which indicates that the time series of acoustic emission waveforms fluctuated more gently. The corresponding Δ*f*(*α*) first increased and then, continuously decreased until it decreased to a minimum point of −1.45 on 3.5, and then showed an increasing trend until it stabilized. During this period, the local stress continued to increase until it broke through the current rock bearing limit. Regarding microfracture penetration at some locations, the location of stress concentration started to shift and strain energy started to be released. With the accumulation of stress, the microfractures all penetrated to form fractures under high stress, resulting in macroscopic damage and release of stress and strain energy, which led to a fall accident.

As shown in Figure 14, it can be seen that there is a close connection between the time-varying response characteristics of the multiple fractal spectrum parameters Δ*α* and Δ*f*(*α*) and the process of rock microfracture development. Before deformation damage occurs in the rock mass, Δ*α* shows a crest trend of increasing and then decreasing, while Δ*f*(*α*) shows a trough trend of decreasing and then increasing. These trend changes can be used as early warning precursor signals of deformation damage. At the time of damage, Δ*α* shows a trend of decreasing and then stable concentration, while Δ*f*(*α*) keeps decreasing until the lowest point. After the damage occurs, both Δ*α* and Δ*f*(*α*) shows a more stable trend.

In particular, when Δ*α* and Δ*f*(*α*) show multiple increases and decreases, this indicates that the stress concentration inside the rock is getting higher and higher, and the strain energy is gathering more and more, which predicts deformation damage of the rock. During this period, reinforcement measures should be taken immediately to control the continued growth of the fissures to prevent greater deformation and destabilization damage in the surrounding rock area.

## 5. Conclusions

In this study, the time series of acoustic emission waveforms were analyzed based on fractal theory. First, wavelet noise reduction method was used to reduce the noise of the obtained waveforms. Subsequently, the fractal dimension curve was adopted to identify the various waveforms generated at the underground mine operation site. Based on the multivariate analysis theory, the multifractal preset parameters of the acoustic emission waveforms were determined, and the precursor information and development law of rock deformation and damage were obtained by analyzing the multifractal time-varying response characteristics. The main conclusions drawn from this study are as follows:(1)The wavelet threshold noise reduction method was used to reduce the noise of the obtained acoustic emission waveform. The analysis results show that the waveform obtained by the wavelet hard threshold noise reduction method has the highest SNR, the lowest RMSE and the best noise reduction performance.(2)The fractal dimensional characteristics of the waveforms were quantified using Hausdorff dimension and box counting dimension, respectively. The results show that the acoustic emission signals generated from different operations downhole had different fractal dimensional performances. In the Hausdorff dimension, the average value of the surrounding rock body waveform was the highest, followed by shovel operation waveform, rock drilling operation waveform and the blasting operation waveform. In the box counting dimension, the fractal dimension of the rock drilling operation waveform was the largest, followed by the surrounding rock body waveform, blasting operation waveform and rock drilling operation waveform. The calculated results obtained by using box counting dimension were clearer and more recognizable, and the box counting dimension could be used as one of the important bases for waveform classification and identification.(3)The key parameters of the multifractal analysis model were determined, and the multifractal characteristics of the rock microfracture waveform time series were characterized based on the parameters Δ*α* and Δ*f*(*α*), when the time length s took the values: *s*_min_ = 870, and *s*_max_ = 934, and the weight factor *q* ∈ [–10, 10].(4)There was a close relationship between the time-varying response characteristics of the multiple fractal spectral parameters of the acoustic emission waveform of the rock body in the quarry area and the sprouting and development of microfracture of the rock body. Before deformation and damage, Δ*α* increased and then decreased, and Δ*f*(*α*) decreased and then increased. At the time of damage, Δ*α* decreased and then stabilized, while Δ*f*(*α*) decreased until its lowest point, then increased and reached a stable state.

## Figures and Tables

**Figure 1 sensors-22-02746-f001:**
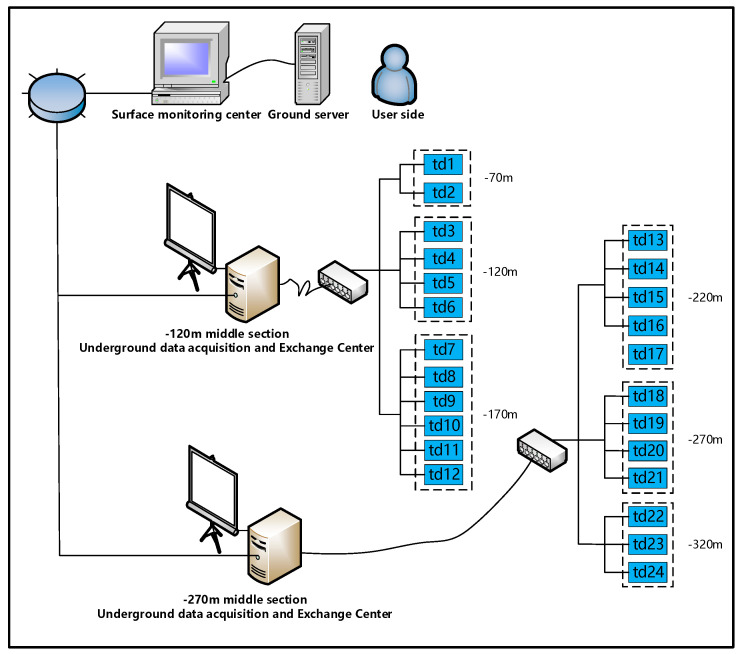
Framework diagram of mine acoustic emission remote monitoring system. (−120 m represents 120 m below the local datum level of the mine and −270 m represents 270 m below the local datum level of the mine.).

**Figure 2 sensors-22-02746-f002:**
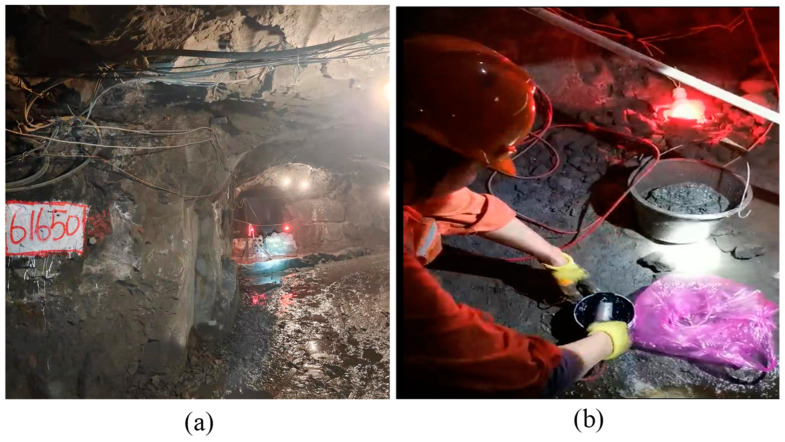
Line installation and layout. (**a**) Optical fiber transmission line erection; (**b**) installation of monitoring probe.

**Figure 3 sensors-22-02746-f003:**
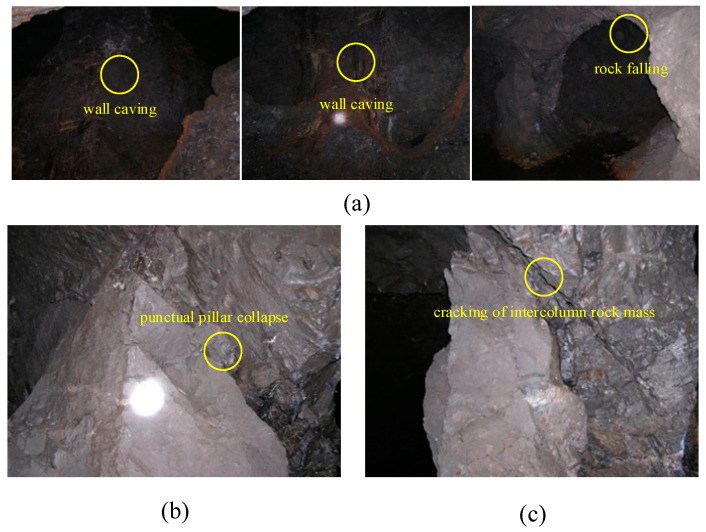
Underground instability and failure phenomena of the mine. (**a**) Underground surrounding rock caving and caving; (**b**)—70 m horizontal 306 stope point column falling across; (**c**)—120 middle section 410 and 412 stope column rock mass cracking.

**Figure 4 sensors-22-02746-f004:**
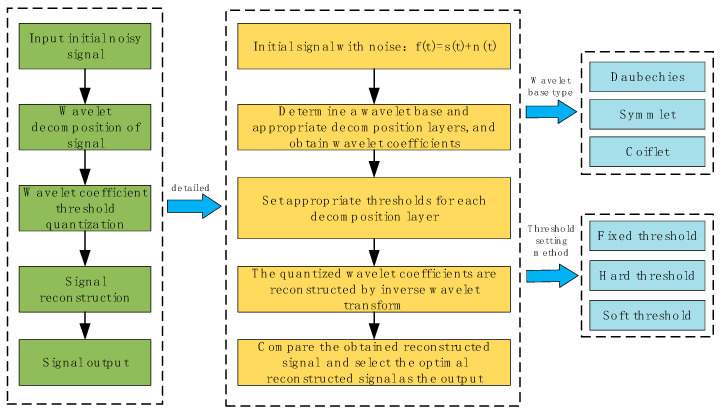
Basic flow of wavelet denoising method.

**Figure 5 sensors-22-02746-f005:**
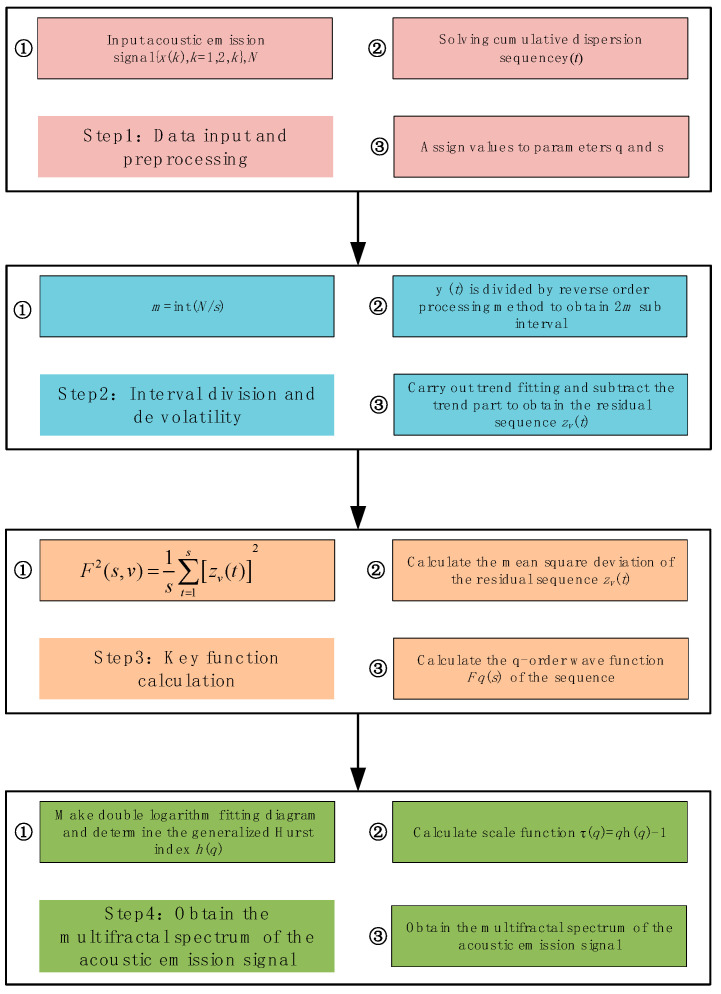
MF-DFA calculation flow chart.

**Figure 6 sensors-22-02746-f006:**
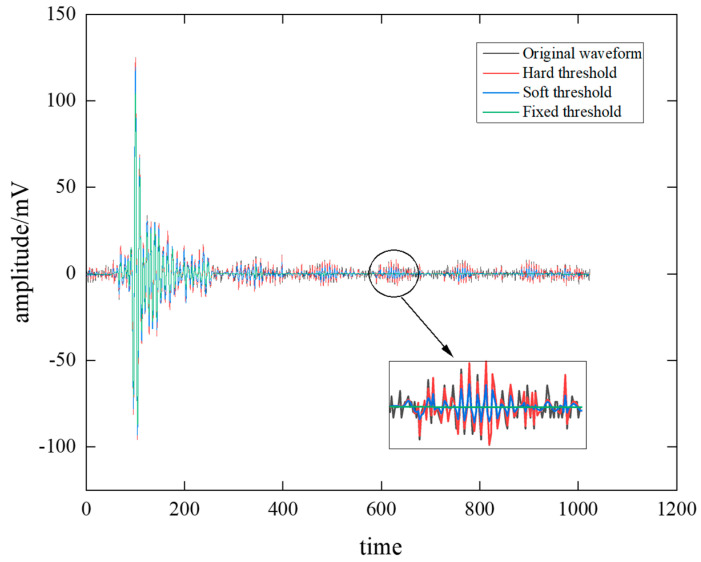
Acoustic emission original waveform of signal A and wavelet denoising.

**Figure 7 sensors-22-02746-f007:**
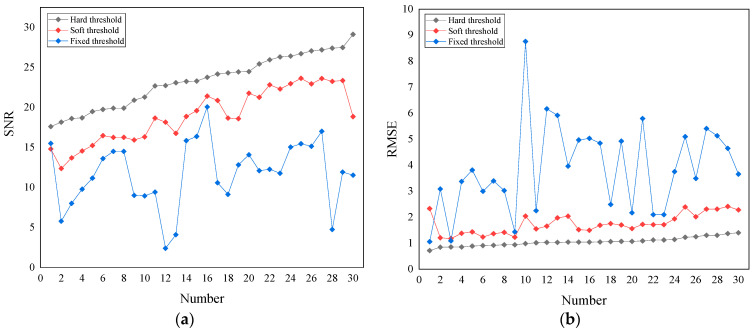
Evaluating indicator performance comparisons with different noise reduction methods: (**a**) SNR, (**b**) RMSE.

**Figure 8 sensors-22-02746-f008:**
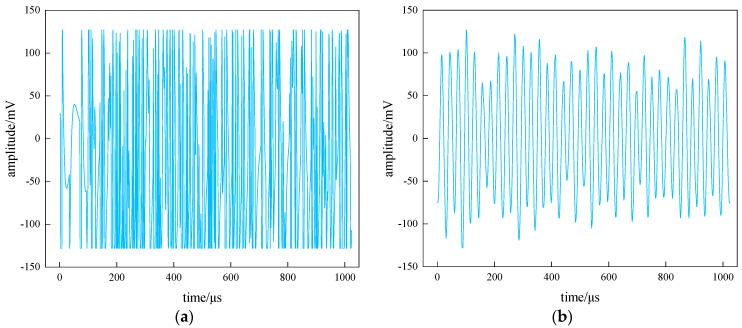
Four typical acoustic emission signal waveforms. (**a**) Blasting operation waveform, (**b**) shovel operation waveform, (**c**) rock drilling operation waveform and (**d**) surrounding rock body waveform.

**Figure 9 sensors-22-02746-f009:**
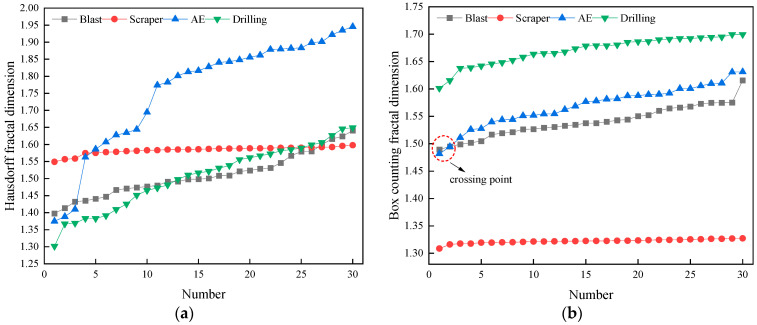
The fractal dimension trend of four waveforms: (**a**) Hausdorff dimension, (**b**) box counting dimension.

**Figure 10 sensors-22-02746-f010:**
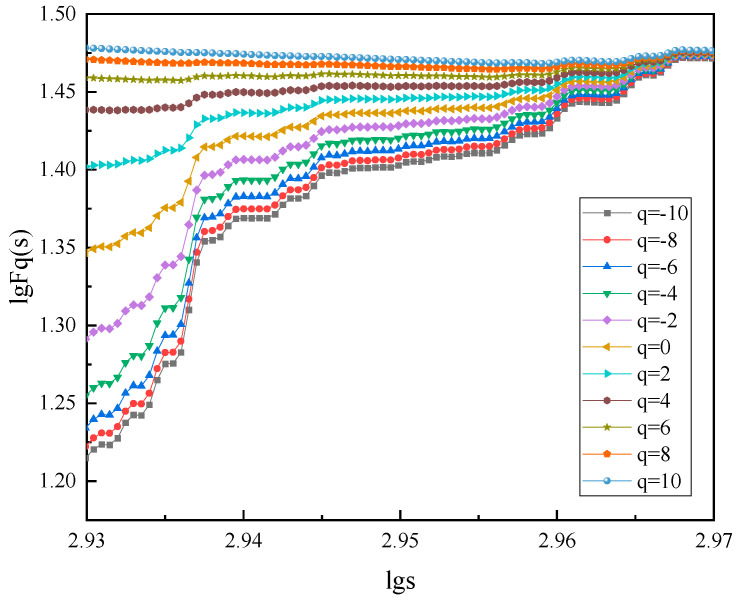
Double logarithm fitting trend of q-order wave function *Fq*(*s*)-*s*.

**Figure 11 sensors-22-02746-f011:**
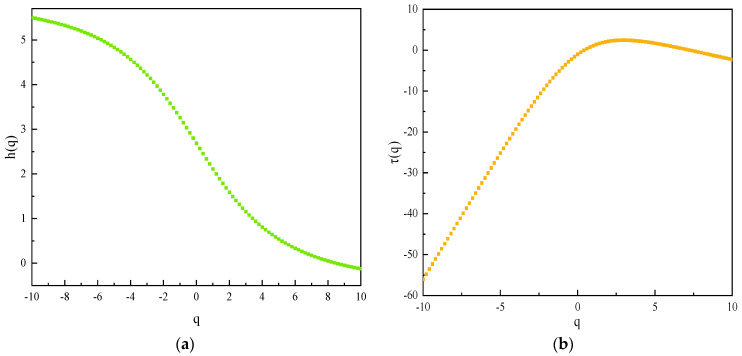
Generalized Hurst exponent (**a**) and scaling function (**b**).

**Figure 12 sensors-22-02746-f012:**
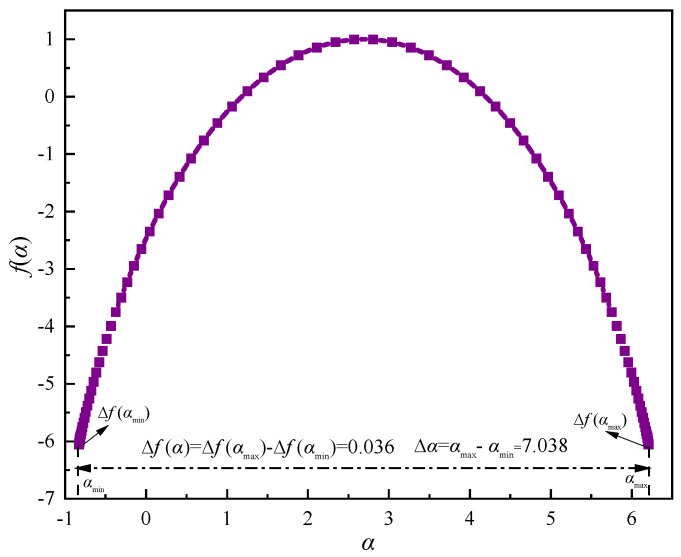
Multifractal spectrum of typical acoustic emission signals.

**Figure 13 sensors-22-02746-f013:**
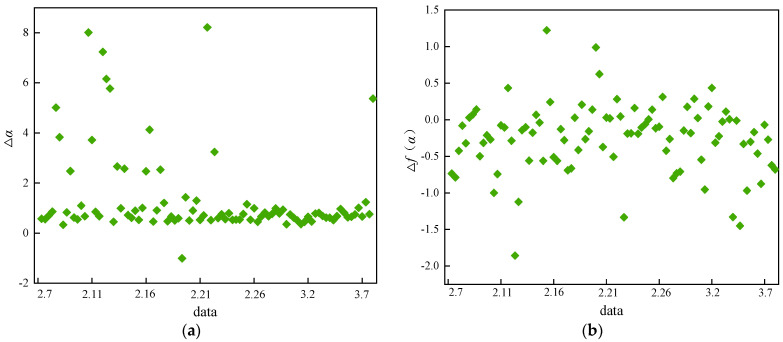
Time varying response law of multifractal characteristic quantity: (**a**) parameter Δ*α*, (**b**) parameter Δ*f*(*α*).

**Figure 14 sensors-22-02746-f014:**
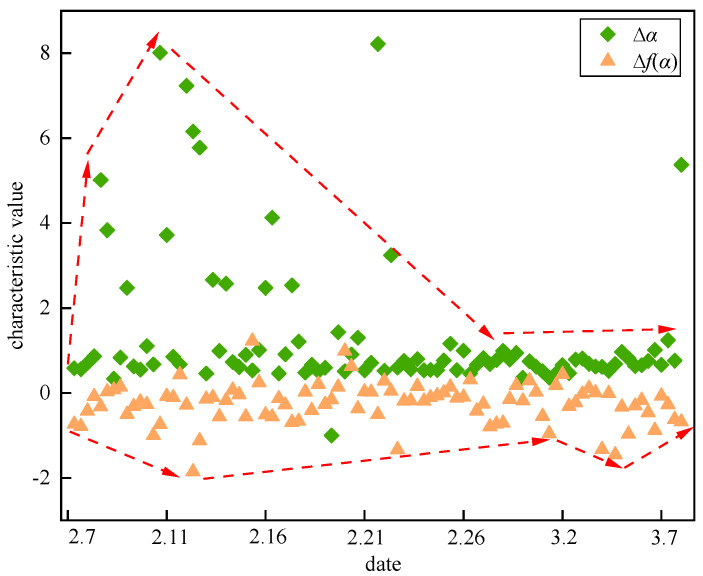
Multifractal development and variation law of rock mass in goaf.

**Table 1 sensors-22-02746-t001:** Comparison of time-frequency domain signal processing technology.

Methods	Strength	Weakness
Short time Fourier transform	Good locality in time domain	Susceptible to analysis window functions
Wigner-Ville distribution	High time-frequency resolution and good time-frequency aggregation performance	Problem of cross interference
Wavelet transform theory	Ensure the authenticity of the signal	Wavelet basis function limited selection
EMD methods	Adaptive time-frequency decomposition	Mode aliasing and endpoint effect

**Table 2 sensors-22-02746-t002:** Parameter characteristics of different types of acoustic emission waveforms.

Signal Type	Attenuation	Frequency/Hz	Energy
Surrounding rock body waveform	Slow	20~80	50~128
Rock drilling waveform	Relatively fast	70~200	120~128
Shovel operation waveform	Relatively fast	100~150	80~110
Blasting operation waveform	Fast	10~300	60~128

## Data Availability

Not applicable.

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
