# Peer review of "Study on Multiple Fractal Analysis and Response Characteristics of Acoustic Emission Signals from Goaf Rock Bodies"

_sensors, 2022, doi:10.3390/s22072746_

Round 1

Reviewer 1 Report

The manuscript presents an informative work of real monitoring work on a mining field by damage-induced acoustic emission with advanced signal processes. The acoustic emission data was processed by incomparable signal processing methods. The work is appreciable for the community by reporting a practical work with multiple processing methods. However, there were serval points needs to be stated clearly. The reviewer lists the detail comments below.

  1. The novelty of the work needs to be clearly state.
  2. The performance of the selected signal process was illustrated. The reviewer suggests the comparison of the selected methods with the conventional methods is better to be included.
  3. The discussion of distinguish between Kaiser effect acoustic emission from the rock and vibration-induced resonance acoustic emission.
  4. The discussion of relation between Kaiser effect acoustic emission waveform between the rock types is better to include. In mine, variance of rock types can have Kaiser effect acoustic emission waveform due to the very different microstructures.
  5. The acoustic signal collection method was not clearly stated. Air-coupled receiving and soil-coupled receiving makes significant difference in the signal-to-noise ratio.
  6. The manuscript needs to be double-checked. For example, Kaiser effect of metal was introduced to the community in 1950 instead of 1905.

Reviewer 2 Report

Xie et al. has investigated the multifractal spectrum of goaf rock acoustic emission signals using fractal theory (MF-DFA). The presented study is well described and up-to-date. The content of the manuscript is according to the journal's expectations. However, a few following suggestions can be incorporated in the revised version. 

Line No 14: Avoid writing a symbol like 'Delta Alpha' in the abstract section. These parameters might not be understood by the reader at the very beginning of the manuscript. Instead, write a little description of this variable.

Line No 45: Change the format of the referred author's name by the Last name of the first author et al. (if more than one author), otherwise the Last name of a single author. For example, Liu et al. [7]......
I can see a similar kind of pattern on the next page(s). Change other sentences accordingly.

Line no: 46: AE?

Line No 117: Cite a reference that describes the hole retention method.

Line No 137: Describe the sign '-' somewhere in Figure 1. -120, .... might create confusion.

Figure 7: What were the possible reason behind a lower amplitude in the case of the surrounding rock body?

Reviewer 3 Report

The manuscript entitled “Study on multiple fractal analysis and response characteristics of acoustic emission signals from goaf rock bodies”, that was submitted to journal “Sensors” of MDPI, deals with the study of time series of acoustic emission waveforms adopting the fractal theory.  

Generally speaking, the content of the manuscript fits in the scope of the journal. The manuscript provides some findings into the AE signal processing.

The presentation of the method is detailed enough.

 Certain comments to the authors follow:

  1.     What is the main problem that prompted you to undertake this study? The aim of the study is missing. Please add additional relative information within the manuscript.
  2.      In Section 2.2 the authors discuss the placement of sensors and optical fibers. A relative photo should be added.
  3.      In Figure 7, four typical acoustic emission signal waveforms (concerning the amplitude) are presented. Please check whether the respective unit is mV and not aJ. In any case please clarify.
  4.      Similarly, as above, please add the unit of the amplitude of the signal in Figure 5.
  5.      The authors should explain and justify their conclusions mentioned in the lines 586-589 of the manuscript.
  6.      It is believed that, for completeness reasons, the reference list should be enriched, taking into account and commenting appropriately, recent papers related to the content of their study from the practical application point of view, as follows:

Fractal characteristics of acoustic emissions from coal under multi-stage true-triaxial compression, Journal of Geophysics and Engineering, 15(5), 2021–2032, 2018. https://doi.org/10.1088/1742-2140/aac31a
Analysis of rupture mode and acoustic emission characteristic of rock and coal samples with holes, Journal of Geophysics and Engineering 16, 811–820, 2019, doi:10.1093/jge/gxz011·      

Round 2

Reviewer 1 Report

The authors addressed the comments well. The reviewer suggests acceptance of the manuscript. 

Reviewer 2 Report

The authors have incorporated the suggestions and given appropriate reasoning for my queries. The revised manuscript can proceed to the next stage.

Thanks!